# Validity and precision of the International Physical Activity Questionnaire for climacteric women using computational intelligence techniques

Ronilson Ferreira Freitas[1], Josiane Santos Brant Rocha[1,2], Laercio Ives Santos[1], André Luiz de Carvalho Braule Pinto[3], Maria Helena Rodrigues Moreira[4], Fernanda Piana Santos Lima de Oliveira[2‡]*, Maria Suzana Marques[1,2‡], Geraldo Edson Souza Guerra Júnior[2‡], Kelma Dayana de Oliveira Silva Guerra[1‡], Andreia Maria Araújo Drummond[5‡], João Victor Villas Boas Spelta[6‡], Carolina Ananias Meira Trovão[2‡], Dorothéa Schmidt França[1,2‡], Lanuza Borges Oliveira[1,2‡], Antônio Prates Caldeira[1,2], Marcos Flávio Silveira Vasconcelos D'Angelo[1]

**1** State University of Montes Claros, Montes Claros, Minas Gerais, Brazil, **2** Fipmoc University Center, Montes Claros, Minas Gerais, Brazil, **3** University of São Paulo, Ribeirão Preto, São Paulo, Brazil, **4** University of Trás-dos-Montes and Alto Douro, Vila Real, Portugal, **5** Federal University of Minas Gerais, Belo Horizonte, Brazil, **6** Federal University of Rio Grande do Norte, Natal, Brazil

☯ These authors contributed equally to this work.
‡ These authors also contributed equally to this work.
* fernandapiana@gmail.com

## Abstract

This study aimed to evaluate the validity and precision of the International Physical Activity Questionnaire (IPAQ) for climacteric women using computational intelligence techniques. The instrument was applied to 873 women aged between 40 and 65 years. Considering the proposal to regroup the set of data related to the level of physical activity of climacteric women using the IPAQ, we used 2 algorithms: Kohonen and k-means, and, to evaluate the validity of these clusters, 3 indexes were used: Silhouette, PBM and Dunn. The questionnaire was tested for validity (factor analysis) and precision (Cronbach's alpha). The Random Forests technique was used to assess the importance of the variables that make up the IPAQ. To classify these variables, we used 3 algorithms: Suport Vector Machine, Artificial Neural Network and Decision Tree. The results of the tests to evaluate the clusters suggested that what is recommended for IPAQ, when applied to climacteric women, is to categorize the results into two groups. The factor analysis resulted in three factors, with factor 1 being composed of variables 3 to 6; factor 2 for variables 7 and 8; and factor 3 for variables 1 and 2. Regarding the reliability estimate, the results of the standardized Cronbach's alpha test showed values between 0.63 to 0.85, being considered acceptable for the construction of the construct. In the test of importance of the variables that make up the instrument, the results showed that variables 1 and 8 presented a lesser degree of importance and by the analysis of Accuracy, Recall, Precision and area under the ROC curve, there was no variation when the results were analyzed with all IPAQ variables but variables 1 and 8. Through

**Data Availability Statement:** All relevant data are within the paper and its Supporting Information files.

**Funding:** The authors received no specific funding for this work.

**Competing interests:** The authors have declared that no competing interests exist.

this analysis, we concluded that the IPAQ, short version, has adequate measurement properties for the investigated population.

## Introduction

Climacteric is a natural phase that women experience during the aging process, and includes the transition between the ovarian reproductive phase and senescence, occurring spontaneously or secondarily to other conditions [1, 2]. This period is marked by a decline in the production of sex hormones, such as estrogen, which can cause physical symptoms, such as: hot flashes and night sweats, urogenital atrophy, sexual dysfunction, mood changes, bone loss and metabolic changes that predispose to cardiovascular diseases and diabetes [1].

In addition to the common changes faced at this stage, due to hypoestrogenism, the climacteric experience is individual and varies for each woman [3]. Age at which menopause occurs, healthy habits, well-being and environment in which they reside are factors that can influence this experience [2]. In addition, women experience physical and psychological changes often associated with aging [3]. The management options for these experiences range from clinical assessment to lifestyle interventions, such as regular physical activity, considered a non-pharmacological intervention, which can minimize the deleterious symptoms resulting from climacteric [4].

Regular physical activity increases bone mineral density, $VO_{2max}$, muscle strength and balance, with a positive impact on body composition [4, 5]. As for clinical factors, it improves the immune system, promoting anti-inflammatory effects [6], as it reduces the risk of insulin resistance, type 2 diabetes, metabolic syndrome and the risk of cardiovascular diseases [7–10]. In addition, it reduces vasomotor symptoms [11] and psychological symptoms, e.g. insomnia, depression and anxiety [11–13], directly impacting the life quality of climacteric women [14].

In view of these findings, it is possible to observe the interest of researchers from Brazil [15–17] and the world [18–21] to assess the level of physical activity in different population groups. Thus, methods have been developed and adapted, as well as used to assess specific health outcomes [15–21]. However, information obtained from the instruments can be divergent, since the populations have specificities and vary according to sex, age, social and cultural aspects and even the individual's cognitive development [22]. In this context, it is important to collect information about the validity and precision of the instrument for the specificities of the population in which it should be used [23].

Regarding the questionnaire options available to assess the level of physical activity, the most used instrument is the International Physical Activity Questionnaire (IPAQ) [24], due to its practicality and low cost of application to a greater number of people [25]. However, despite the literature showing indications related to the validity and reproducibility of the IPAQ in the young [23] and adult [15, 26, 27] Brazilian population, no studies with these characteristics were found involving the population of climacteric women. Thus, this study aimed to assess the validity and precision of the IPAQ for climacteric women using computational intelligence techniques.

In this study, 2 clustering algorithms were used: k-means [28] and Kohonen [29], and 3 cluster validation indexes: Silhouette [30], PBM [31] and the Dunn index [32], to determine the validity, and what would be the best number of categories that the set of data related to the level of physical activity of climacteric women is divided, using the IPAQ. The Exploratory Factor Analysis (EFA) technique was used to assess the construct's validity through the analytical factor approach [33, 34] and the Random Forests (RF) classification technique, to measure

the importance of the variables that make up the IPAQ [35]. To classify these variables, three algorithms were used: Support Vector Machine (SVM) [36], Artificial Neural Network (ANN) [37] and Decision Tree (DT) [38]. In addition, we use 4 metrics to assess the quality of the results: Accuracy, Recall, Precision and area under the ROC curve (AUC) [39].

## Methods

This study is derived from the research project entitled "Health conditions of climacteric women: an epidemiological study", carried out in the city of Montes Claros, Minas Gerais, Brazil, by a group of researchers. This project considers the general health of climacteric women in this important transition phase to be its central theme. As it involves human beings, this study was submitted, evaluated and approved for execution by the Research Ethics Committee of the Fipmoc University Center under opinion No. 817,666/2014.

### Participants

The study was carried out in Montes Claros, Minas Gerais, Brazil, from August 2014 to August 2015, whose target population was composed of 30,801 climacteric women registered in 73 units of Family Health Strategies (FHS), which represents the Primary Health Care (PHC) mechanism in the public health system in Brazil [40].

The sampling was of the probabilistic type and the sample selection occurred in two stages. Each FHS team was taken as a conglomerate, with 20 units drawn, covering the urban and rural areas for data collection. Then, a proportional number of women was randomly selected, according to the climacteric stratification criteria (pre, peri and post-menopause), of the Brazilian Climacteric Society [41]. For each unit, 48 women were selected, making a total of 960 women summoned. To incorporate the structure of the complex sample plan in the statistical analysis of the data, each respondent was associated with a weight $w$, which corresponded to the inverse of their probability of inclusion in the sample ($f$) [42].

Women aged between 40 and 65 years were considered eligible to participate in the research. They registered with the selected FHS teams, with physical and psychological conditions to answer the questionnaires. Pregnant, postpartum and bedridden women were not included.

### Instruments

The research used a structured questionnaire that included the following variables: sociodemographic (age, education, type of school attended, paid work and family income) to characterize the sample profile, and the International Physical Activity Questionnaire (IPAQ) [24] in order to assess the practice of physical activity.

### International Physical Activity Questionnaire (IPAQ)

The basic instrument of this study is the International Physical Activity Questionnaire (IPAQ), an instrument proposed by the International Group for Consensus on Physical Activity Measures, constituted under the seal of the World Health Organization, with representatives from 25 countries, including Brazil. It is an instrument developed with the purpose of estimating the level of habitual practice of physical activity for the population between 18 and 65 years old from different countries and different socio-cultural backgrounds [24].

The IPAQ has been adapted for several languages, with two versions available, one in the long format and the other in the short format. Both versions are self-applicable or can be applied in an interview format. In addition, they seek to assess the frequency and duration of

the walks, as well as the daily activities that require physical efforts of moderate and vigorous intensity, having as reference period a typical week or the last week before the data collection period [23].

For the present study, the short version of the IPAQ was used, as it is the version most frequently suggested for use in both national [43–45] and international [21, 46] studies with different populations. This version consists of eight variables related to physical activity performed in the last week, shown in Table 1.

## Procedures

Initially, training was provided for data collectors and interviewers. The entire process was supervised by the research coordinator. Then, a pilot study was carried out in an FHS unit, with women belonging to the age group studied and who were not part of the final sample. The pilot study allowed the questionnaire and the interviewers' performance to be tested in practice. The field research started with the selection of women who were invited to participate in the research on a previously established date. The final sample, considering the missing data, without compromising the minimum required sample size, was 873 climacteric women, who signed the Informed Consent Form.

## Statistical data analysis

**Reclustering and evaluation of validation indexes.**   With regard to clustering and the evaluation of data validation indexes, cluster analysis has been used in several real problems. Dividing objects, beings or instances into groups is a task that the human being can perform without much effort. However, when the number of instances is large, this task becomes a complex problem, and the use of computerized methods is necessary. Clustering consists of dividing n instances of data into a k number of clusters, so that instances of the same cluster are more similar than instances of different clusters [47].

By using different heuristics to generate the cluster, each method can generate different results for the same data set. Thus, researchers have developed techniques that help to measure

**Table 1. Variables that make up the International Physical Activity Questionnaire (IPAQ).**

| Variable | Assessed Aspect |
|---|---|
| 1 | How many days of the week have you **WALKED** for at least 10 continuous minutes at home or at work, as a form of transportation to get from one place to another, for leisure, for pleasure or as a form of exercise? |
| 2 | On days when you **WALKED** for at least 10 continuous minutes, how much time in total did you spend walking **each day**? |
| 3 | On how many days in the last week, have you performed **MODERATE** activities for at least 10 continuous minutes, such as: cycling lightly on the bicycle; swimming; dancing; doing light aerobics; playing recreational volleyball; carrying light weights; doing chores in the house, in the yard or in the garden, such as sweeping, vacuuming, gardening; or any activity that **moderately** increased your breathing or heart rate? |
| 4 | On the days you did these **MODERATE** activities, for at least 10 continuous minutes, how much time in total did you spend doing them **each day**? |
| 5 | On how many days in the past week, have you performed **VIGOROUS** activities for at least 10 continuous minutes, such as: running; doing aerobic gymnastics; playing soccer; cycling fast on the bicycle; playing basketball; doing heavy chores in the house, in the yard or digging the garden; carrying heavy weights; or any activity that made your breathing or heartbeat increase **VERY MUCH**? |
| 6 | On the days you did these **VIGOROUS** activities, for at least 10 continuous minutes, how much time in total did you spend doing them **each day**? |
| 7 | How much time in total do you spend **sitting** on **a weekday**? |
| 8 | How much time in total do you spend **sitting** on **a weekend day**? |

quantitatively how good a given cluster is. Thus, the cluster validation indexes appear. These indexes judge statistically and based on a value the quality of the clusters found. In general, the more compact the groups formed, the better the result of the evaluation of the indexes [48].

In this study, we used 2 methods of clustering data. The first method was k-means. This method searches for a set of k vectors to represent k groups, as follows: k vectors or centers are initialized randomly; then, each training instance is associated with the most similar center; each vector is recalculated using the average of the instances associated with it; each training instance is associated again with the most similar center and recalculation is performed; the process ends when all instances of iteration t+1 belong to the same group of iteration t [28].

The second clustering method was developed by Kohonen [29]. It is a competitive neural network composed of two inputs: one composed of the input instances and the other composed of weight vectors, which must be adjusted during learning. During training, the instances are presented to the learning algorithm, the neurons compete with each other and the weights of the winning neuron are updated according to the Eq (1).

$$W_t = W_t + \partial(X_i - W_{t-1})$$ (1)

In (1): W represents the weight vector of the winning neuron, X the current instance, and $\partial$ the learning rate that decreases during the execution of the algorithm. In competition, a similarity metric is used and the winning neuron will be the most similar in relation to the training instance. The method is iterative and ends according to some criteria, such as number of times or when the weights of all neurons stabilize. The final weight vectors of each neuron are used as prototypes of the clusters formed by the training instances.

We also used 3 cluster assessment metrics described below. The main objective of this stage was to determine what would be the best number of categories in which the set of data related to the level of physical activity of climacteric women using the IPAQ is divided.

The Silhouette index measures the quality of the cluster based on the proximity between the instances of the same cluster and the distance of instances of a cluster to the nearest one. Also the higher its value, the better the cluster. In this way, it is possible to determine the best number of clusters [30].

To obtain the index value, the silhouette of each instance must first be calculated using Eq (2):

$$I_{SIL} = \frac{b(i) - a(i)}{max(a(i), b(i))}$$ (2)

In it: a(i) is the average distance from instance i to all other instances in its cluster and b(i) is the minimum distance from instance i to all other instances that do not belong to its cluster.

The index is calculated for each instance separately, the value for a cluster is the average of the index of all instances in it, and the index for the clustering will be the average of the indexes for all clusters.

Another index used in this study was the PBM, obtained using the distances between the elements of the cluster, as well as their centers and their distances between the centers of each cluster [31].

$D_c$ is the greatest distance between two centers, given by Eq (3).

$$D_C = \max_{k<k\prime} d(G^k, G^{k\prime})$$ (3)

On the other hand, $E_w$ denotes the sum of the distances from the points of each cluster to its center Eq (4); and $E_t$ is the sum of the distances from all points to the G center of the entire

data set Eq (5):

$$E_w = \sum_{k=1}^{K}\sum_{i\in I_k} d(M_i, G^k) \tag{4}$$

$$E_t = \sum_{i=1}^{N} d(M_i, G) \tag{5}$$

PBM is given by:

$$PBM = \left(\frac{1}{K} \times \frac{E_t}{E_w} \times D_C\right)^2 \tag{6}$$

As with the previous index, the best value for PBM is the highest.

The third index used was Dunn, which is measured by the ratio of separation within and between clusters. The original Dunn can be calculated by Eq (7), where: $dist(C_i, C_j)$ is a function of similarity between clusters i and j defined by Eq (8); and $diam(C_g)$ is the dispersion of cluster g given by Eq (7). The higher the index value the better the clustering, so Dunn can be used to identify the ideal number of clusters, where the k with the highest index value is the ideal amount [32].

$$Dunn(k) = \min_{i=1,\ldots,k}\left\{\min_{i=1,\ldots,k}\left\{\frac{dist(C_i, C_j)}{\max_{g=1,\ldots,k} diam(C_g)}\right\}\right\} \tag{7}$$

$$dist(C_i, C_j) = \min_{x\in C_i, y\in C_j} d(x, y) \tag{8}$$

$$diam(C_g) = \max_{x,y\in C_g} d(x, y) \tag{9}$$

The *ClusterCrit* package of Software R, version 3.4.2, was used to carry out the clustering experiments, as well as to organize the validation indexes.

## Concordance and reliability test

The construct's validity was evaluated through the factor analytical approach using the Exploratory Factor Analysis (EFA) technique. For the extraction of the factors, the technique by main components with rotation using the Varimax orthogonal method was applied. The Kaiser-Meyer-Olkin (KMO) and Bartlett's Sphericity tests were performed to verify the fit of the data to the EFA. The KMO aimed to verify if the individuals who participated in the response to the instrument did so consistently. If the KMO value is greater than 0.60, the responses are considered consistent. In construct validation by factor analysis, Bartlett's sphericity test must be statistically significant (p < 0.05) [33, 34].

Precision was assessed using standardized Cronbach's alpha (α) internal consistency, which is the most used method in cross-sectional studies—here measurements are performed in just a single moment [49] and the metrics used are presented in different scales (seven days for variables 1, 3 and 5; minutes for variables 2, 4, 6, 7 and 8) [50]. This coefficient allows to identify the internal consistency of the test, that is, the coherence between each test variable [51]. According to Pasquali [52], Cronbach's alpha coefficient varies between 0.00 (lack of reliability) and 1.00 (perfect reliability). In this validation, the standardized alpha value ≥ 0.60 was considered acceptable for the assessment of the construct (group of questions) [53].

Cronbach's alpha coefficient was calculated with the aid of the software *Statistical Package for Social Sciences* (SPSS)®, version 21.

## Measurement of the importance of the variables that make up the IPAQ

In order to measure the importance of the variables that make up the IPAQ, the Random Forests (RF) classification technique was used. It is a technique based on decision trees that uses a set of trees to perform the classification. Each tree in the set is induced from randomly selected instances and variables and, for a classification problem, the prediction of the model is determined by the majority vote, that is, the most prevalent class among the classes predicted by the set of trees. In addition to the prediction, the RF can list the variables in order of capacity or predictive importance and this importance can be used to select variables as inputs for other classification models. To measure the importance of a variable m, the RF adds the impurity (Gini index) in all the nodes of a tree. Then, the values of m are shuffled randomly between the instances and the sum of the impurities is performed again. The importance of variable m is given by the average decrease in impurity among all trees [35].

The RF was calculated with the aid of the Matlab R2015b software to measure the importance of the variables.

## Classification algorithms of the variables that make up the IPAQ

Classification is a Machine Learning technique, which aims to assign predefined categories or classes to data instances [54]; its application takes place in two stages. The first, called training —the model is built to describe a predetermined set of classes. In this stage, a function is built that discriminates the stages of the addressed problem. In the second stage, the constructed model is used to classify a different set of instances than the set used in the first stage [54]. In this study, the Support Vector Machines (SVM) [36] classification models were used; Artificial Neural Networks (ANN) [37] and Decision Trees (DT) [38] to classify the variables that make up the IPAQ.

SVM aims to draw a hyperplane, in order to maximize the distance between instances of two different classes. When the data of the problem in question has only two characteristics, this hyperplane is represented by a line, in a data set with n characteristics, a hyperplane with n dimensions is necessary to adapt to the data [36].

An Artificial Neural Network is a computational method that tries to simulate the way a human brain learns. A biological neuron receives input information from an external source and combines these inputs with non-linear operations to produce results based on the assimilated knowledge. The basic processing unit of an ANN is the artificial neuron, which, similar to the biological one, communicates through a large number of connections forming a weighted network, in which the input signals are sent to other neurons. Neurons are variables of interconnected processors, operating in parallel to perform a certain task. In general, ANNs are composed of layers organized by a defined number of neurons and in order for them to perform the proposed tasks, they must undergo a learning process. This process consists, in short, in finding values of synaptic weights that best associate input elements with output elements (interest). Therefore, an ANN can be defined as a computational model of biological inspiration defined to process neurons and connections between them with weights linked to them [37].

A Decision Tree is a data structure defined recursively and composed of internal nodes (decision nodes) and leaf nodes. An internal node contains a test on some attribute and for each test result there is an edge for a subtree. A leaf node corresponds to a class in classification problems or a probability in regression problems. There are several methods of inducing DT. In this work, we used the Classification and Regression Tree (CART) method because it presents several advantages over other methods, such as noise robustness, low computational cost and the ability to deal with redundant attributes [38].

The experiments with the SVM, ANN and DT methods were performed using the Matlab R2015b software for data classification. We used 4 metrics to evaluate the quality of the results: Accuracy, Recall, Precision and area under the ROC curve (AUC) [39]. For all methods, a 5-folder cross-validation format was used, in which 3 folders were used for training, 1 folder for parameter calibration, and the other folder to test the model and the average value of each metric.

## Results

This study have the participation of 873 women, with a mean age of $51.04 \pm 7.1$ years. As for education, 5.8% of the women were illiterate, 35.2% had attended only primary education, 26.5% attended elementary school II, 26.9% attended high school, and only 5.6% attended higher education, with 97.3% attending public schools. It was observed that 59.6% of women reported not working and 63.8% reported that the family income is up to 1 minimum wage.

### Reclustering and evaluation of validation indexes

Considering the proposal of regrouping and validation, to measure the quality of the formed clusters, three indexes were used: in Silhouette, the k-means and Kohonen algorithms presented an index of 0.38 for the clustering of the sample in two clusters; in the PBM index, the Kohonen and k-means algorithms showed an index of 0.35 and 0.34, respectively, for clustering into two clusters; in the Dunn index, both the k-means and Kohonen algorithms showed better indexes for clustering into four clusters, as it can be seen in Table 2.

### Concordance and reliability test

It was tested whether the correlation matrix was adequate for the factorial analysis procedures. $KMO = 0.76$ and the test of Bartllet $X^2$ (8) = 21800; p <0.001 indicated the adequacy of the data. A factor analysis using the maximum likelihood method, with varimax rotation, was conducted. The scree plot exam showed a three-factor solution, confirmed by a parallel analysis (Fig 1).

As we can see in Table 3, factor 1 was composed of variables 3 to 6, with an explained variance of 27% and internal consistency of 0.81, through Cronbach's alpha. The second factor, composed of variables 7 and 8, explained 19% of the observed variance, with Cronbach's alpha of 0.85. Finally, factor 3 carried variables 1 and 2, explaining 13% of the variance, with

**Table 2. Regrouping and evaluation of the International Physical Activity Questionnaire (IPAQ) validation indexes for climacteric women.**

| Number of Groups | Silhouette | PBM | Dunn |
|---|---|---|---|
| k-means | | | |
| 2 | 0,389 | 0,351 | 0,072 |
| 3 | 0,337 | 0,339 | 0,076 |
| 4 | 0,252 | 0,149 | 0,082 |
| 5 | 0,252 | 0,206 | 0,033 |
| 6 | 0,278 | 0,183 | 0,037 |
| Kohonen | | | |
| 2 | 0,386 | 0,348 | 0,075 |
| 3 | 0,337 | 0,341 | 0,076 |
| 4 | 0,301 | 0,326 | 0,084 |
| 5 | 0,280 | 0,248 | 0,029 |
| 6 | 0,277 | 0,164 | 0,043 |

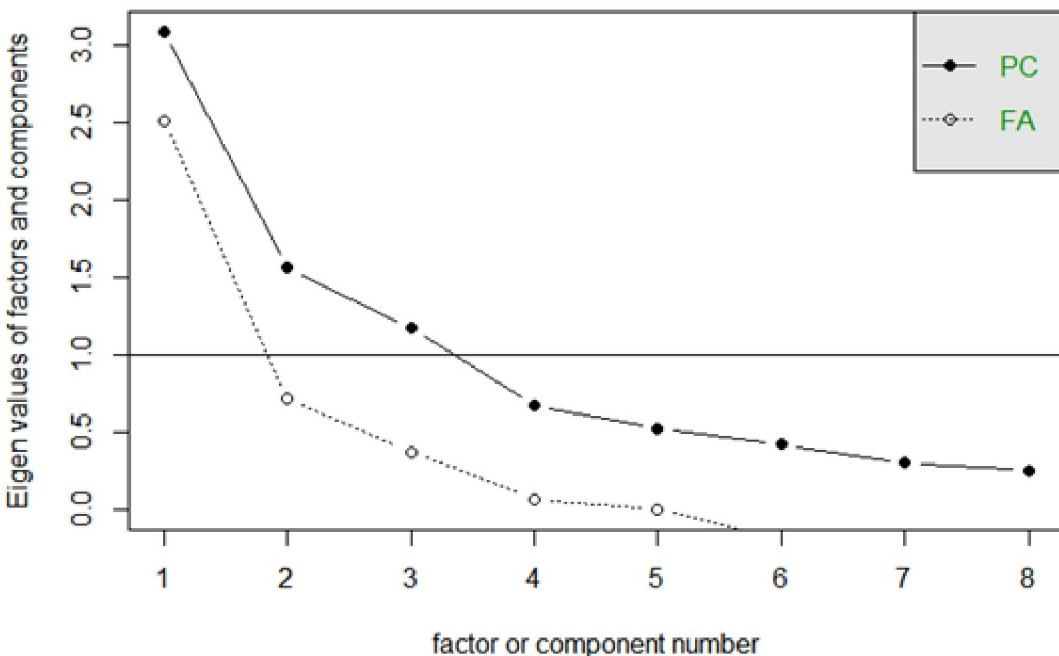

**Fig 1. Scree plot para o International Physical Activity Questionnaire (IPAQ).**

Cronbach's alpha of 0.63. These results indicate that a 3-factor solution showed good internal consistency and explained about 59% of the variance, pointing to the adequacy of the IPAQ in terms of its validity and precision for climacteric women.

## Measurement of the importance of the variables that make up the IPAQ

The Random Forests technique was used to assess the order of capacity or importance of the variables that make up the International Physical Activity Questionnaire (IPAQ). The results showed that variables 1 and 8, presented a lesser degree of importance, as shown in Fig 2.

**Table 3. Factor loads of the variables that make up the International Physical Activity Questionnaire (IPAQ).**

| Variables | Factor 1 | Factor 2 | Factor 3 | h2 | u2 | com |
|---|---|---|---|---|---|---|
| 1 | | | 0,55 | 0,40 | 0,60 | 1,6 |
| 2 | | | 0,81 | 0,66 | 0,34 | 1,0 |
| 3 | 0,78 | | | 0,64 | 0,36 | 1,1 |
| 4 | 0,72 | | | 0,54 | 0,46 | 1,0 |
| 5 | 0,57 | | | 0,35 | 0,65 | 1,2 |
| 6 | 0,75 | | | 0,61 | 0,39 | 1,2 |
| 7 | | 0,86 | | 0,76 | 0,24 | 1,1 |
| 8 | | 0,85 | | 0,73 | 0,27 | 1,0 |
| Own Values | 2,16 | 1,50 | 1,03 | | | |
| % Variance Explained | 27,0 | 19,0 | 13,0 | | | |
| Cronbach's Alpha* | 0,81 | 0,85 | 0,63 | | | |

* Standardized Cronbach's Alpha.

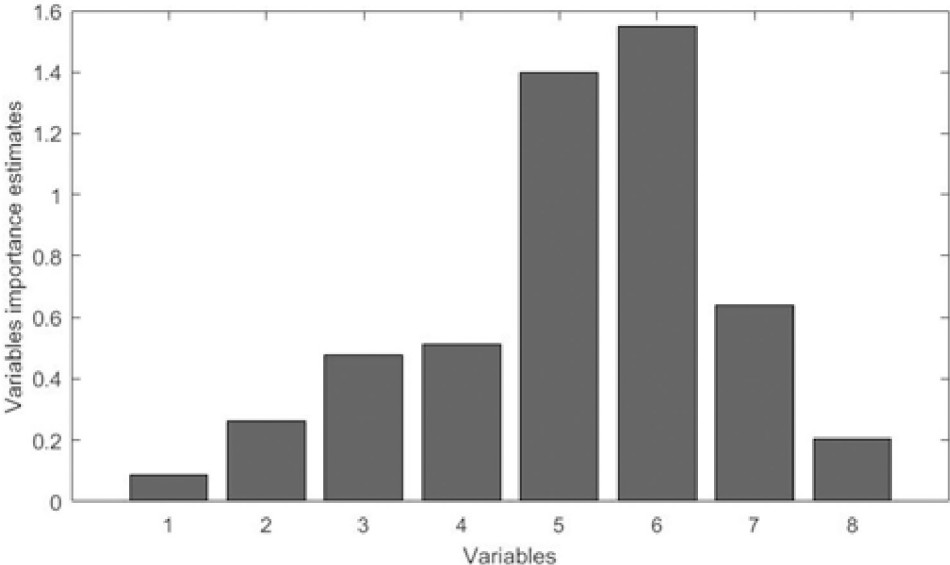

**Fig 2. Importance of the variables that make up the International Physical Activity Questionnaire (IPAQ).**

## Classification algorithms of the variables that make up the IPAQ

To classify the variables that make up the IPAQ, the Support Vector Machine (SVM), Artificial Neural Network (ANN) and Decision Tree (DT) were used. Table 4 shows the results of the experiment, in which the entire base was used to train and test the model. In the SVM, ANN and DT tests, by analyzing the Accuracy, Recall, Precision and area under the ROC curve, there was no variation when the results were analyzed with all the variables that make up the IPAQ and without variables 1 and 8.

Table 5 shows the results of the 5-fold cross-validation experiment. In the SVM, RNA and DT tests, by analyzing the Accuracy, Recall, Precision and area under the ROC curve, the results reinforce the findings of the experiment, in which the entire base was used to train and

**Table 4. Experiment using the entire base to train and test the model.**

|  | Accuracy | Recall | Precision | AUC |
|---|---|---|---|---|
| SVM | 0,966 | 0,886 | 0,863 | 0,923 |
| SVM–without variable 1 | 0,965 | 0,886 | 0,855 | 0,919 |
| SVM–without variable 8 | 0,965 | 0,903 | 0,844 | 0,914 |
| SVM–without variables 1 and 8 | 0,965 | 0,903 | 0,844 | 0,914 |
| ANN | 0,977 | 0,921 | 0,905 | 0,946 |
| ANN–without variable 1 | 0,975 | 0,924 | 0,893 | 0,940 |
| ANN–without variable 8 | 0,972 | 0,907 | 0,886 | 0,936 |
| ANN–without variables 1 and 8 | 0,974 | 0,914 | 0,886 | 0,936 |
| DT | 0,979 | 0,912 | 0,928 | 0,957 |
| DT–without variable 1 | 0,964 | 0,938 | 0,816 | 0,903 |
| DT–without variable 8 | 0,964 | 0,938 | 0,816 | 0,903 |
| DT–without variables 1 and 8 | 0,964 | 0,938 | 0,816 | 0,903 |

SVM = Support Vector Machine; ANN = Artificial Neural Network; DT = Decision Tree; AUC = Area Under the ROC Curve.

**Table 5. 5-fold cross-validation experiment.**

|  | Accuracy | Recall | Precision | AUC |
|---|---|---|---|---|
| SVM | 0,957 | 0,841 | 0,849 | 0,912 |
| SVM–without variable 1 | 0,961 | 0,868 | 0,851 | 0,915 |
| SVM–without variable 8 | 0,958 | 0,840 | 0,860 | 0,918 |
| SVM–without variables 1 and 8 | 0,964 | 0,840 | 0,860 | 0,918 |
| ANN | 0,948 | 0,785 | 0,830 | 0,899 |
| ANN–without variable 1 | 0,951 | 0,811 | 0,832 | 0,902 |
| ANN–without variable 8 | 0,951 | 0,811 | 0,835 | 0,903 |
| ANN–without variables 1 and 8 | 0,958 | 0,853 | 0,839 | 0,908 |
| DT | 0,937 | 0,913 | 0,701 | 0,843 |
| DT–without variable 1 | 0,941 | 0,913 | 0,731 | 0,858 |
| DT–without variable 8 | 0,937 | 0,913 | 0,701 | 0,843 |
| DT–without variables 1 and 8 | 0,941 | 0,913 | 0,731 | 0,858 |

SVM = Support Vector Machine; ANN = Artificial Neural Network; DT = Decision Tree; AUC = Area Under the ROC Curve.

test the model, where no variation was observed when the results were analyzed with all the variables of the IPAQ and without variables 1 and 8.

## Discussion

In Brazil, the instruments that dominate epidemiological surveys associated with physical activity are questionnaires. Despite the problems related to the subjective format of the evaluation and the estimation errors, these instruments are important for data collection due to their ease of application, their low cost, the great population applicability and for allowing to know, for example, the level of physical activity in specific populations, using fewer financial resources, when compared to other instruments for measuring the level of physical activity [23]. In the case of women in the climacteric period, phase of life in which there is a decrease in the production of sex hormones, consequently impacting the level of physical activity [55], questionnaires represent the most accessible instrument for the assessment of habitual physical activity, especially in studies of an epidemiological nature [23].

In Brazil, research has validated the IPAQ for specific populations, which allowed the conduct of epidemiological studies that assess the level of physical activity in teenagers [23], adults [26, 27, 56] and elderly people [57, 58]. These validation studies for specific populations are important due to the characteristics of the information they propose to observe for each investigated group. When the instrument is not validated for these populations, it can generate inconsistencies in the results, when compared with specific instruments, which may demonstrate limitations regarding the criteria of validity and reliability of the results [23]. However, based on the findings of national and international literature, this seems to be the first study that sought to analyze the validity and precision of IPAQ for climacteric women, which may contribute to the assessment of the level of physical activity, considering the specific characteristics of this population.

In the present study, considering the results of the reclustering analysis using the k-means and Kohhonen algorithms, in the Silhouette and PBM indexes, it was observed that the recommended for the IPAQ, when applied to climacteric women, is to categorize the results in two groups (sufficiently active and insufficiently active). However, the Dunn index suggests categorization into four groups, which is already proposed in the literature by Matsudo et al. [26], which recommends classifying the investigated as sedentary, insufficiently active, active and

very active. It should be noted that other studies carried out by Brazilian researchers have already done the clustering into two groups [59–63]. In it, was possible to observe the reclustering between the sedentary/insufficiently active and active/very active categories. However, these studies did not present analytical tests to prove this reclustering, as stated in the present study, which used computational intelligence techniques to evaluate and validate this action.

In the analysis of the construct's validity using exploratory factor analysis, the factors extracted were: walking (factor 3), moderate and vigorous physical activity (factor 1) and physical inactivity (factor 2). Based on the structure of the instrument, according to which variables 1 and 2 refer to the practice of walking, variables 3 to 6 to moderate and vigorous physical activity and variables 7 and 8 to the time that the individual remains seated, these three factors were already expected, since the classification of the level of physical activity takes into account the frequency, duration and intensity of the activities carried out during the week prior to the interview, including the time they remained seated during one day of the week and of the weekend [23].

Regarding the internal consistency of the questionnaire treated in the present study, Cronbach's alpha values were found to vary between 0.63 to 0.85, being considered acceptable for the construction of the construct, pointing out that the IPAQ version met the proposed acceptability criteria ($\alpha > 0,60$). Valim et al. [64], claim that a Cronbach's alpha of 0.63 has moderate reliability. However, the results for factors 1 and 2, which had an alpha coefficient $> 0.80$, were similar to those reported in a literature review on internal consistency of the IPAQ, in which the reliability results for seven self-report measures of physical activity evaluated in adults showed reliability correlations ranging from 0.34 to 0.89, with a median of about 0.80 [65].

In this context, taking into account that the construct validity has the objective of sustaining the instrument's ability to measure what it is designed to measure [64], the reliability of the IPAQ was evidenced to measure the practice of physical activity by climacteric women, in the Brazilian context. It is important to highlight that the value of Cronbach's alpha is influenced both by the value of the correlations of the variables and by the number of variables evaluated. Therefore, factors with few variables tend to have smaller Cronbach alphas, while a matrix with high inter-correlations tends to have a high alpha value [66].

Through the analysis, it is possible to state that the IPAQ has adequate measurement properties for climacteric women, at least as good as other instruments used. The results of this study were supported by another study carried out to assess the reliability of IPAQ in several countries, in addition to assessing the suitability of this instrument to determine the level of physical activity in the population. Satisfactory monitoring measurement properties were observed among adults 18 to 65 years of age, acceptable in different contexts [24]. Its short version is already recommended for national monitoring of the general population [26].

To measure the importance of the variables that make up the IPAQ, the Random Forests technique was used, which demonstrated that variables 1 and 8 were less important. In addition, in the classification of the variables that make up the IPAQ using the Support Vector Machine (SVM), the Artificial Neural Network (ANN) and the Decision Tree (DT), both in tests in which the entire base was used to train and test the model, as in the 5-fold cross-validation experiment by analyzing the accuracy, recall, precision and area under the ROC curve (AUC), there was no variation when the results were analyzed with all IPAQ variables and without variables 1 and 8, which mathematically suggests the removal of these variables.

Variable 1 refers to the number of days of the week that the individual walked for at least 10 continuous minutes at home or at work, as a form of transportation to go from one place to another, for leisure, for pleasure or as a way of exercise. Considering the findings of the present study, it is reinforced that in addition to the number of days, the time and intensity of physical exercise must also be taken into consideration to estimate the level of physical activity [67],

since the efficiency of regular exercise is determined by the combination of frequency, intensity and duration to obtain a training effect [68]. In this context, considering the Compendium of Physical Activity (CPA) [69] (developed by researchers at Stanford University, in the United States, to be used in epidemiological studies, where the intensities of each type of exercise were standardized, seeking to facilitate the coding of physical activity obtained in research) it was suggested that walking should be considered a vigorous activity.

Thus, the importance of the practice of physical activity is reinforced, which mobilizes large muscle groups, maintained continuously, of a rhythmic and aerobic nature, in order to reduce the positive energy balance, related to the unregulated energy intake and physical inactivity, increasing the risk for cardiovascular disease [70].

Furthermore, it is suggested that these findings are related to the limitations of IPAQ reported by Matsudo et al. [26], that there is a difficulty for those evaluated to estimate, quantify and accurately determine what would be an ordinary week, in the case of the practice of moderate activity, and the total time spent sitting, during a weekend day.

It is important to note that the values reported in the IPAQ consider not only physical activity developed during leisure, but also consider the practice at work and domestic services, as well as commuting. Although the positive aspect of such a clustering allows incorporating different dimensions in which physical activity can be developed, the impossibility of evaluating each one in isolation in the short version of the questionnaire imposes limits on data analysis [60]. The authors Hallal et al. [71] report that validation studies in Latin America suggest that IPAQ has high reliability and moderate criteria validity compared to other instruments that assess the level of physical activity in the population. However, cognitive interviews suggested that occupational and domestic sections cause confusion among respondents, and as the short version of the instrument completely considers the domains of physical activity, people tend to provide inaccurate answers.

Regarding variable 8, which investigates how much time the individual spends sitting on a weekend day, a Brazilian study conducted by Matsudo et al. [26], in order to determine the validity of the IPAQ in a sample of Brazilian adults, he concluded that this variable would be optional, since the sitting activities should be asked preferably during the week, as they are more representative on these days than on the weekend [26]. This reinforces the findings of this study with respect to the less importance of this variable to estimate the activities seated during a weekend day.

As a limitation, this study highlights the lack of comparison of the results obtained by the IPAQ short version with those of other instruments for assessing the level of physical activity in climacteric women, due to the lack of validation studies of instruments for the evaluation of this variable in this population. Thus, investigations regarding the criterion validity and discriminating validity of the IPAQ were made impossible. Therefore, it is suggested that this theme be addressed in future works in order to compare the validity and reliability of the IPAQ, with other instruments.

## Conclusion

The psychometric properties of IPAQ were highlighted, configuring this instrument as a potential tool to assess the level of activity in climacteric women. Through the tests to evaluate the clusters using the Silhouette and PBM indexes, it was observed that the recommended for the IPAQ, when applied to climacteric women, is to categorize the results in two groups (sufficiently active and insufficiently active).

The results show that the instrument has reliability and validity for this specific population. The classification of the items that make up the IPAQ, using the SVM, ANN and DT

algorithms, points out that the values of Accuracy, Recall, Precision and area under the ROC curve did not differ with the removal of variables 1 and 8.

## Supporting information

**S1 Database.**
(SAV)

## Author Contributions

**Conceptualization:** Ronilson Ferreira Freitas, Josiane Santos Brant Rocha, Maria Helena Rodrigues Moreira, Antônio Prates Caldeira, Marcos Flávio Silveira Vasconcelos D'Angelo.

**Data curation:** Ronilson Ferreira Freitas, Josiane Santos Brant Rocha, Marcos Flávio Silveira Vasconcelos D'Angelo.

**Formal analysis:** Ronilson Ferreira Freitas, Josiane Santos Brant Rocha, Laercio Ives Santos, André Luiz de Carvalho Braule Pinto, Maria Helena Rodrigues Moreira, Fernanda Piana Santos Lima de Oliveira, Maria Suzana Marques, Antônio Prates Caldeira, Marcos Flávio Silveira Vasconcelos D'Angelo.

**Investigation:** Ronilson Ferreira Freitas, Maria Suzana Marques.

**Methodology:** Ronilson Ferreira Freitas, Josiane Santos Brant Rocha, Laercio Ives Santos, André Luiz de Carvalho Braule Pinto, Maria Suzana Marques, Marcos Flávio Silveira Vasconcelos D'Angelo.

**Project administration:** Josiane Santos Brant Rocha.

**Resources:** Josiane Santos Brant Rocha.

**Software:** Laercio Ives Santos, André Luiz de Carvalho Braule Pinto, João Victor Villas Boas Spelta, Marcos Flávio Silveira Vasconcelos D'Angelo.

**Supervision:** Josiane Santos Brant Rocha, Marcos Flávio Silveira Vasconcelos D'Angelo.

**Validation:** Ronilson Ferreira Freitas, Josiane Santos Brant Rocha, Dorothéa Schmidt França, Marcos Flávio Silveira Vasconcelos D'Angelo.

**Visualization:** Josiane Santos Brant Rocha, Laercio Ives Santos, André Luiz de Carvalho Braule Pinto, Maria Helena Rodrigues Moreira, Fernanda Piana Santos Lima de Oliveira, Geraldo Edson Souza Guerra Júnior, Kelma Dayana de Oliveira Silva Guerra, Andreia Maria Araújo Drummond, João Victor Villas Boas Spelta, Carolina Ananias Meira Trovão, Dorothéa Schmidt França, Lanuza Borges Oliveira, Antônio Prates Caldeira, Marcos Flávio Silveira Vasconcelos D'Angelo.

**Writing – original draft:** Ronilson Ferreira Freitas, Josiane Santos Brant Rocha, Laercio Ives Santos, André Luiz de Carvalho Braule Pinto, Maria Helena Rodrigues Moreira, Fernanda Piana Santos Lima de Oliveira, Maria Suzana Marques, Antônio Prates Caldeira, Marcos Flávio Silveira Vasconcelos D'Angelo.

**Writing – review & editing:** Ronilson Ferreira Freitas, Josiane Santos Brant Rocha, Laercio Ives Santos, André Luiz de Carvalho Braule Pinto, Maria Helena Rodrigues Moreira, Fernanda Piana Santos Lima de Oliveira, Maria Suzana Marques, Geraldo Edson Souza Guerra Júnior, Kelma Dayana de Oliveira Silva Guerra, Andreia Maria Araújo Drummond, João Victor Villas Boas Spelta, Carolina Ananias Meira Trovão, Dorothéa Schmidt França,

Lanuza Borges Oliveira, Antônio Prates Caldeira, Marcos Flávio Silveira Vasconcelos D'Angelo.

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
