## [Decision Letter · Decision Letter 0]

26 Oct 2020

PONE-D-20-28544

VALIDITY AND PRECISION OF THE INTERNATIONAL PHYSICAL ACTIVITY QUESTIONNAIRE FOR CLIMACTERIC WOMEN USING COMPUTATIONAL INTELLIGENCE TECHNIQUES

PLOS ONE

Dear Dr. Piana Santos Lima de Oliveira,

Thank you for submitting your manuscript to PLOS ONE. After careful consideration, we feel that it has merit but does not fully meet PLOS ONE’s publication criteria as it currently stands. Therefore, we invite you to submit a revised version of the manuscript that addresses the points raised during the review process.

We look forward to receiving your revised manuscript.

Kind regards,

Seyedali Mirjalili

Academic Editor

PLOS ONE

Journal Requirements:

2. Please include your tables as part of your main manuscript and remove the individual files. Please note that supplementary tables should remain as separate "supporting information" files.

Reviewers' comments:

Reviewer's Responses to Questions

**Comments to the Author**

1. Is the manuscript technically sound, and do the data support the conclusions?

Reviewer #1: Yes

Reviewer #2: Partly

2. Has the statistical analysis been performed appropriately and rigorously? 

Reviewer #1: No

Reviewer #2: No

3. Have the authors made all data underlying the findings in their manuscript fully available?

Reviewer #1: Yes

Reviewer #2: Yes

4. Is the manuscript presented in an intelligible fashion and written in standard English?

Reviewer #1: Yes

Reviewer #2: Yes

5. Review Comments to the Author

Reviewer #1: In this paper, the authors apply the computational methods to investigate the precision of IPAQ for climacteric women. Overall, this paper is well organized. There are several minor and major observations listed as follows:

1. Lack of literature review, the authors are suggested to discuss more recent and related works.

2. The main contributions of this paper is not clear. Please point out the main contributions in the last paragraph of Introduction.

3. Authors are encouraged to provide a flow-diagram of the proposed work.

4. Random forest is a well regard classifier. The authors are suggested to apply it not only in measuring the important of the variable but also in classification stage.

5. The parameter settings of machine learning algorithms such as number of splits in DT, number of neurons, hidden layers in ANN, the kernel function used in SVM need to be provided.

6. Instead of using machine learning algorithms, deep learning methods such as convolutional neural network can be also applied for classification.

7. Authors should perform the statistical analysis (e.g. Friedman test or Anova test or Wilcoxon test) to support the classification results.

Reviewer #2: This is a very interesting topic, but a number of a major and minor amendments are required as follows:

* The authors apply Kohonen and k-means algorithms to evaluate the validity of the (IPAQ) for climacteric women, why these algorithms, why not other algorithms? what is your contribution in terms of these computational intelligence?

* The authors did not include related work of this study, existing methods and recent related work should be included and explored.

*For the method, authors should discussed and clearly explain the algorithms they used and show their contribution.

* another concern is that the figures and tables should be placed within the text.

* There is no statistical test to judge about the significance of the method’s results. Without such a statistical test, the conclusion cannot be supported.

6. PLOS authors have the option to publish the peer review history of their article (what does this mean?). If published, this will include your full peer review and any attached files.

Reviewer #1: No

Reviewer #2: No

---

## [Author Response · Author response to Decision Letter 0]

28 Nov 2020

Dear Dr. Seyedali Mirjalili

Academic Editor, Plos One Magazine

We initially record our thanks to the considerations on the article PONE-D-20-28544, entitled "Validity and precision of the international physical activity questionnaire for climacteric women using computational intelligence techniques" and to the suggestions presented. Below, we present our responses to each of the items scored, as a way to facilitate the review of the same and we are available for any clarifications or suggestions for improving the work.

Reviewer 1

COMMENTS

1. Lack of literature review, the authors are suggested to discuss more recent and related works.

RESPONSE: A literature review was carried out on the use of computational intelligence techniques for the selection of variables in epidemiological studies and for the validation of data collection instruments, which justifies the use of computational intelligence techniques for the validation of IPAQ for women weather, which was included in the introduction of the manuscript section.

2. The main contributions of this paper is not clear. Please point out the main contributions in the last paragraph of Introduction.

RESPONSE: As the last paragraph of the introduction of the manuscript, as suggested by the reviewer, we include the main contributions of this work.

3. Authors are encouraged to provide a flow-diagram of the proposed work.

RESPONSE: We created a flowchart (Figure 1), which was inserted in the introduction of the manuscript, showing all stages of validation and evaluation of the accuracy of the IPAQ.

4. Random forest is a well regard classifier. The authors are suggested to apply it not only in measuring the important of the variable but also in classification stage.

RESPONSE: Suggestion accepted. Random forest was used to classify the variables that make up the IPAQ. It was included in the classification methodology and results.

5. The parameter settings of machine learning algorithms such as number of splits in DT, number of neurons, hidden layers in ANN, the kernel function used in SVM need to be provided.

RESPONSE: The machine learning algorithm parameter settings, such as number of divisions in DT, number of neurons, hidden layers in ANN, the kernel function used in SVM were provided in the work methodology (Table 1), as suggested.

6. Instead of using machine learning algorithms, deep learning methods such as convolutional neural network can be also applied for classification.

RESPONSE: Yes, we agree with the reviewer, however, deep Convolutional RNAs need a large volume of characteristics and training examples to be efficient [1,2] which can be provided by complex data such as images, texts and videos and, therefore, are more indicated for these types of data. The data set used in this study has only 8 characteristics and 873 examples and, therefore, we consider it small for learning via Convolutional RNA. In addition, the purpose of using the Machine Learning methods for classification was to verify whether the elimination of variables 1 and 8 would affect the structure of the IPAQ and for that we used classification techniques widely used in the literature.

[1] RGB-D object recognition and pose estimation based on characteristics of pre-trained convolutional neural networks

[2] SALAMON, Justin; BELLO, Juan Pablo. Deep convolutional neural networks and increased data for classification of environmental sounds. IEEE Signal Processing Letters, v. 24, n. 3, p. 279-283, 2017.

7. Authors should perform the statistical analysis (e.g. Friedman test or Anova test or Wilcoxon test) to support the classification results.

RESPONSE: We agree with the reviewer. The Kruskal-Wallis test was used to compare the results. The data from the statistical test of the cross-validation is in the table referring to the cross-validation (Table 5). The statistical test data of everyone trained and everyone tested cannot be done because there is only a single execution of the algorithm for each configuration.

Reviewer 2

COMMENTS

1. The authors apply Kohonen and k-means algorithms to evaluate the validity of the (IPAQ) for climacteric women, why these algorithms, why not other algorithms? what is your contribution in terms of these computational intelligence?

RESPONSE: KMeans was used because it is a popular method of data partitioning widely used in many fields, including data mining, pattern recognition, decision support and machine learning and also due to its ability to handle numerical variables and usage of a spacing heuristic to choose the initial groups in order to avoid suboptimal solutions. Kohonen RNA was used to reinforce kMeans' findings regarding optimal grouping, since both are partition based grouping methods but with different grouping heruistics. The two methods are quite consolidated in the architecture and meet the first objective of the article, which is to verify the number of categories that IPAQ is divided into. We have included this information in the manuscript methodology.

2. The authors did not include related work of this study, existing methods and recent related work should be included and explored.

RESPONSE: A literature review was carried out on the use of computational intelligence techniques for the selection of variables in epidemiological studies and for the validation of data collection instruments, which justifies the use of computational intelligence techniques for the validation of IPAQ for women weather, which was included in the introduction of the manuscript.

3. For the method, authors should discussed and clearly explain the algorithms they used and show their contribution.

RESPONSE: The machine learning algorithm parameter settings, such as number of divisions in DT, number of neurons, hidden layers in ANN, the kernel function used in SVM were provided in the work methodology, as suggested (Table 1).

4. Another concern is that the figures and tables should be placed within the text.

RESPONSE: Figures and tables have been included in the body of the text, as recommended.

5. There is no statistical test to judge about the significance of the method’s results. Without such a statistical test, the conclusion cannot be supported.

RESPONSE: We agree with the reviewer. The Kruskal-Wallis test was used to compare the results. The data from the statistical test of the cross-validation is in the table referring to the cross-validation (Table 5). The statistical test data of everyone trained and everyone tested cannot be done because there is only a single execution of the algorithm for each configuration.

---

## [Decision Letter · Decision Letter 1]

9 Dec 2020

PONE-D-20-28544R1

VALIDITY AND PRECISION OF THE INTERNATIONAL PHYSICAL ACTIVITY QUESTIONNAIRE FOR CLIMACTERIC WOMEN USING COMPUTATIONAL INTELLIGENCE TECHNIQUES

PLOS ONE

Dear Dr. Piana Santos Lima de Oliveira,

Thank you for submitting your manuscript to PLOS ONE. After careful consideration, we feel that it has merit but does not fully meet PLOS ONE’s publication criteria as it currently stands. Therefore, we invite you to submit a revised version of the manuscript that addresses the points raised during the review process.

We look forward to receiving your revised manuscript.

Kind regards,

Seyedali Mirjalili

Academic Editor

PLOS ONE

Reviewers' comments:

Reviewer's Responses to Questions

**Comments to the Author**

1. If the authors have adequately addressed your comments raised in a previous round of review and you feel that this manuscript is now acceptable for publication, you may indicate that here to bypass the “Comments to the Author” section, enter your conflict of interest statement in the “Confidential to Editor” section, and submit your "Accept" recommendation.

Reviewer #1: (No Response)

Reviewer #2: All comments have been addressed

2. Is the manuscript technically sound, and do the data support the conclusions?

Reviewer #1: Yes

Reviewer #2: Yes

3. Has the statistical analysis been performed appropriately and rigorously? 

Reviewer #1: N/A

Reviewer #2: Yes

4. Have the authors made all data underlying the findings in their manuscript fully available?

Reviewer #1: Yes

Reviewer #2: Yes

5. Is the manuscript presented in an intelligible fashion and written in standard English?

Reviewer #1: Yes

Reviewer #2: Yes

6. Review Comments to the Author

Reviewer #1: In the revised paper, the authors have addressed most of my concerns. I have one minor observation.

1. In Table 5, please provide the detail how the p-value is calculated. Also, the p-value was very high, a discussion on this finding is required.

Reviewer #2: Well done, the authors did great effort in response to the most comments given in first round review, however still some minor corrections to be made to enhance the manuscript further as follows:

1-Figure 1, the caption should be placed below the figure. Also, here as suggested in previous review to include the flowchart of the proposed method and the description of the proposed methods in methodology part.

2- The authors recommended to reorganized the method, so they include first the proposed method mention in 1. then (Participants, Instruments, International Physical Activity Questionnaire (IPAQ) and Procedures) put all of them in one section named e.g. (Area of the study).

4. Its recommended to follow the journal format and numbering of sections.

3- The authors used Kruskal-Wallis Test which is good, however they didn't explain within the text which method is significant using this test. Include the discussion of the use of the Kruskal-Wallis Test.

7. PLOS authors have the option to publish the peer review history of their article (what does this mean?). If published, this will include your full peer review and any attached files.

Reviewer #1: No

Reviewer #2: No

---

## [Author Response · Author response to Decision Letter 1]

18 Dec 2020

Dear Dr. Seyedali Mirjalili

Academic Editor, Plos One Magazine

We initially record our thanks to the considerations on the article PONE-D-20-28544, entitled "Validity and precision of the international physical activity questionnaire for climacteric women using computational intelligence techniques" and to the suggestions presented. Below, we present our responses to each of the items scored, as a way to facilitate the review of the same and we are available for any clarifications or suggestions for improving the work. We are left with a little doubt with the question number 4 of the reviewer 2, about the numbering of the sessions. We did not find this information in the magazine's formatting guidelines, but as he suggested, we did so in the body of the text. 

Reviewer 1

Comments

1. In Table 5, please provide the detail how the p-value is calculated. Also, the p-value was very high, a discussion on this finding is required.

Response: Methods were included in the session as the p-value was calculated, and we inserted information about the p-value as a caption in Table 5. In this study we adopted a significance level of 0.05, that is, for values less than or equal to 0.05 we reject the null hypothesis (which says that among the groups evaluated all values are equal) and for values greater than 0.05 we accept the null hypothesis. That is, when the p-value is greater than 0.05 it means that all values are equal, therefore, for values considered low or high, the level of significance was respected. The Kruskal-Wallis test is a non-parametric method that uses the difference between the averages of the stations in each group to determine the p-value, in this study this difference is small for most cases of comparison and a small difference contributes to a high p-value. For example, in the precision of the SVM method, the posts in each group had averages of: 9, 10.9, 10 and 12.1 which generated a p-value of 0.859. The recall of the SVM had rank averages of: 9.9, 10.5, 10.6 and 11 and this generated a p-value of 0.993.

Reviewer 2

Comments

1-Figure 1, the caption should be placed below the figure. Also, here as suggested in previous review to include the flowchart of the proposed method and the description of the proposed methods in methodology part.

Response: We included a caption below the figure and inserted the flowchart in the methods section, as suggested by the reviewer.

2- The authors recommended to reorganized the method, so they include first the proposed method mention in 1. then (Participants, Instruments, International Physical Activity Questionnaire (IPAQ) and Procedures) put all of them in one section named e.g. (Area of the study).

Response: We accepted the reviewer's suggestion, and the participating sessions, instruments, International Physical Activity Questionnaire (IPAQ) and procedures were included in a single session called study procedures.

3- The authors used Kruskal-Wallis Test which is good, however they didn't explain within the text which method is significant using this test. Include the discussion of the use of the Kruskal-Wallis Test.

Response: Methods were included in the session as the p-value was calculated, and we inserted information about the p-value as a caption in Table 5. In this study we adopted a significance level of 0.05, that is, for values less than or equal to 0.05 we reject the null hypothesis (which says that among the groups evaluated all values are equal) and for values greater than 0.05 we accept the null hypothesis. That is, when the p-value is greater than 0.05 it means that all values are equal, therefore, for values considered low or high, the level of significance was respected. The Kruskal-Wallis test is a non-parametric method that uses the difference between the averages of the stations in each group to determine the p-value, in this study this difference is small for most cases of comparison and a small difference contributes to a high p-value. For example, in the precision of the SVM method, the posts in each group had averages of: 9, 10.9, 10 and 12.1 which generated a p-value of 0.859. The recall of the SVM had rank averages of: 9.9, 10.5, 10.6 and 11 and this generated a p-value of 0.993.

4. Its recommended to follow the journal format and numbering of sections.

Response: Suggestion accepted, we have numbered the sections of the article.

---

## [Editor Report · Decision Letter 2]

26 Dec 2020

VALIDITY AND PRECISION OF THE INTERNATIONAL PHYSICAL ACTIVITY QUESTIONNAIRE FOR CLIMACTERIC WOMEN USING COMPUTATIONAL INTELLIGENCE TECHNIQUES

PONE-D-20-28544R2

Dear Dr. Piana Santos Lima de Oliveira,

We’re pleased to inform you that your manuscript has been judged scientifically suitable for publication and will be formally accepted for publication once it meets all outstanding technical requirements.

Kind regards,

Seyedali Mirjalili

Academic Editor

PLOS ONE
---

## [Editor Report · Acceptance letter]

4 Jan 2021

PONE-D-20-28544R2 

Validity and Precision of the International Physical Activity Questionnaire for Climacteric Women using Computational Intelligence Techniques 

Dear Dr. Piana Santos Lima de Oliveira:

I'm pleased to inform you that your manuscript has been deemed suitable for publication in PLOS ONE. Congratulations! Your manuscript is now with our production department. 

Kind regards, 

on behalf of

Prof. Seyedali Mirjalili 

Academic Editor

PLOS ONE